# Global use of electronic patient-reported outcome systems in nephrology: a mixed methods study

Nicola Elizabeth Anderson ,[1,2,3] Derek Kyte ,[1,4] Christel McMullan ,[1,5,6] Paul Cockwell,[1,7] Olalekan Lee Aiyegbusi ,[1,3,6,8,9] Rav Verdi,[10] Melanie Calvert [1,3,5,6,8,9]

For numbered affiliations see end of article.

**Correspondence to**
Dr Nicola Elizabeth Anderson;
n.e.anderson.1@bham.ac.uk

## ABSTRACT

**Objectives** The use of electronic patient-reported outcome (ePRO) systems to support the management of patients with chronic kidney disease is increasing. This mixed-methods study aimed to comprehensively identify existing and developing ePRO systems, used in nephrology settings globally, ascertaining key characteristics and factors for successful implementation.

**Study design** ePRO systems and developers were identified through a scoping review of the literature and contact with field experts. Developers were invited to participate in a structured survey, to summarise key system characteristics including: (1) system objectives, (2) population, (3) PRO measures used, (4) level of automation, (5) reporting, (6) integration into workflow and (7) links to electronic health records/national registries. Subsequent semistructured interviews were conducted to explore responses.

**Setting and participants** Eligible systems included those being developed or used in nephrology settings to assess ePROs and summarise results to care providers. System developers included those with a key responsibility for aspects of the design, development or implementation of an eligible system.

**Analytical approach** Structured survey data were summarised using descriptive statistics. Interview transcripts were analysed using Codebook Thematic Analysis using domains from the Consolidated Framework for Implementation Research.

**Results** Fifteen unique ePRO systems were identified across seven countries; 10 system developers completed the structured survey and 7 participated in semistructured interviews. Despite system heterogeneity, reported features required for effective implementation included early and sustained patient involvement, clinician champions and expanding existing electronic platforms to integrate ePROs. Systems demonstrated several common features, with the majority being implemented within research settings, thereby affecting system implementation readiness for real-world application.

**Conclusions** There has been considerable research investment in ePRO systems. The findings of this study outline key system features and factors to support the successful implementation of ePROs in routine kidney care.Cite Now

## STRENGTHS AND LIMITATIONS OF THIS STUDY

⇒ The mixed-methods approach offered a greater understanding of the barriers and facilitators of electronic patient-reported outcome (ePROs) implementation, supplementing survey findings on key system characteristics with in-depth interview data from ePROs system developers.

⇒ The electronic survey and interview topic guide were developed with input from stakeholders, including clinicians, PROs methodologists and patients.

⇒ The review was restricted to English-language publications of ePRO systems. Therefore, it is possible that potentially relevant publications published in other languages were excluded.

⇒ Purposive sampling methods were used due to the limited numbers of people who could act as a primary data source, that is, survey and interview participants could have a clinical or non-clinical role but must have carried key responsibility for aspects of design, development or implementation of an eligible ePRO system.

⇒ It is possible that other ePRO systems that could be used in nephrology settings were not identified by this study.

## INTRODUCTION

Patients at all stages of chronic kidney disease (CKD), and particularly those undertaking kidney replacement therapy, experience a high symptom burden and often report diminished health-related quality of life (HRQOL).[1–3] However, some symptoms remain under-recognised and unalleviated, leading to increased patient burden.[4] HRQOL is not only an important outcome in itself but also associated with clinical outcomes such as healthcare utilisation and mortality.[5] Symptom burden and impact on HRQOL can be assessed using patient-reported outcomes (PROs), defined as 'any report of the status of a patient's health condition that comes directly from the patient, without interpretation of the patient's response by a clinician or anyone'.[6]

PROs assessing the impact of CKD and associated treatments can be collected electronically (electronic PROs; ePROs).

With the increase in telemedicine, accelerated by the COVID-19 pandemic,[7 8] the collection of ePROs has increasing significance. Web-based systems and 'bring your own device' schemes offer patients new ways to report symptoms, function and HRQOL; they can report PROs in 'real-time' from home or in clinic prior to their consultation. Such use could assist in the management of symptoms, while ensuring that healthcare resources are used effectively to maximise impact.[9]

There is increasing evidence, particularly from oncology, that ePROs are cost-effective,[10] can facilitate shared decision-making, promote self-management, improve symptoms and HRQOL, reduce unplanned hospitalisation and enhance long-term outcomes including survival.[11–14] In contrast to paper-based collection, ePRO systems offer enhanced ease of use and reduced burden, increased user satisfaction (patient and clinician) and lower rates of missing dataf.[15–18]

Evidence around implementation of ePROs in nephrology is growing. Studies from North America demonstrate the feasibility of ePRO collection in haemodialysis (HD) settings,[16 19 20] while the Australian Symptom monitoring With Feedback Trial[21] and Canadian EMPATHY studies[22] are cluster randomised controlled trials exploring the usefulness of integrating PRO assessments in the clinical management of patients receiving HD. In advanced CKD populations, the use of an electronic Patient-Reported Outcome Measure in the Management of Patients with Advanced CKD (RePROM) study in the UK piloted the use, and established feasibility, of an ePRO for remote symptom monitoring in real time.[9 23] The AmbuFlex telepro system is used in practice to manage renal outpatient follow-up in Denmark and as a complimentary tool in consultations[24 25] and is currently being evaluated through the PRO-KID trial: a non-inferiority pragmatic randomised controlled trial investigating the effectiveness of the quality of care, use of resources and patient outcomes associated with PRO-based follow-up in patients with CKD.[24]

With this growth of ePRO system research and implementation in CKD globally, there is an opportunity to explore the key characteristics associated with successful implementation to support widespread adoption. The objectives of this study were to[1] comprehensively identify ePRO systems designed specifically for use with CKD populations, including those under development,[2] to explore system characteristics, including methods of administration, levels of integration into existing workflow and electronic health records (EHR) and the reporting of ePROs to manage patient safety.

The aim is being to provide a comprehensive classification of core factors, which contribute to successful implementation of ePROs in nephrology, including common facilitators and barriers.

## STUDY DESIGN
### Methods
Predicated on the research paradigm of pragmatism, with a focus on analysis of study data through the lens of its practical consequences and actionable knowledge,[26] this three-phased study used mixed methods:

Phase 1 comprised a scoping review identifying ePRO systems and corresponding developers.[27] Eligible systems were those being developed, used or under study that were designed for or reconfigured specifically for use in adult nephrology settings but excluding those developed for the management of acute kidney injury and paediatric CKD populations. Systems were also excluded if PROs were not assessed electronically and did not provide a summary of the patients' responses to their care provider for use in routine care, that is, were being used solely to collect research or population-level data or to assess the effectiveness of an intervention.

Eligible systems from any country were included. A system developer could have a clinical or non-clinical role but must have carried key responsibility for aspects of design, development or implementation of an eligible ePRO system.

Systems were identified through publications in English, including conference abstracts and grey literature; adapting a previously applied search strategy is used to investigate nephrology PRO measures.[28] Databases (OVID Medline, EMBASE, APA Psychinfo and CINAHL) were searched from dates of inception to 15 December 2021 (see online supplemental file 1, for example, search strategy) followed by ongoing 'pearl' and 'snowballing' methodology, that is, searching from known key references and checking reference lists.[29–31] Field experts were also consulted. Data from identified publications were used to create a structured survey exploring key system characteristics (see online supplemental file 2), and to identify system developers.

Phase 2 involved survey administration via an online survey platform Smart Survey (www.smartsurvey.co.uk), which was piloted prior to use. System developers were invited to participate via email. The survey focused on system design and software features, integration of e-PRO collection and reporting in clinical care.

Phase 3: optional follow-up semistructured interviews were undertaken online via videoconferencing software with survey participants, or a suitable team member nominated by the original survey respondent. The purpose of these interviews was to expand on survey responses and allow more detailed system/organisational enquiry. All follow-up interviews used a piloted study-specific topic guide (see online supplemental file 3). Participants were given the opportunity to check and review transcripts, to comment and avoid disclosure of any proprietary information. Interviews were conducted by NEA, a renal research nurse with previous experience of conducting qualitative research.[32] This study was undertaken as part of a Doctoral Research Fellowship, which was disclosed in the participant information sheet and during the consent

**Table 1** Overview of ePRO systems (see online supplemental file 4 for additional system data)

| Name of system | Country | Population | In current use yes/no | System ePROs |
|---|---|---|---|---|
| ANZDATA SWIFT PROMs Module[21 58 75–77] | Australia | In-centre HD | Yes—pilot study reported, Cluster RCT in progress | SONGHD fatigue measure IPOSRenal EQ5D5L |
| cPRO-Collaborate Kidney Care[40] | USA | CKD Stages 13 CKD Stages 45 pre-dialysis | Yes—clinical demonstration model | PROMIS and FACIT measures |
| Cambian (ePRO-KIDNEY)[16 17 42 78–81] | Canada | Home HD, PD | No—platform used within reported research projects | Patient Assessment of Care for Chronic Conditions20 questionnaire (PACIC20) Kidney Disease Quality of Life36 (KDQOL36) Edmonton Symptom Assessment System renal (ESASr), EQ5D5L |
| OPT-ePRO[68 82–84] | UK | CKD Stages 13, CKD Stages 45 pre-dialysis, In-centre HD, Home HD, PD transplant, conservative care | No—pilot study reported | POS-S-RENAL, EQ-5D-5L |
| Penguin (Cievert Ltd) (no published data) | UK | Transplant | Yes—research pilot in progress | New study-specific measure |
| RePROM[9 23 85–87] | UK | CKD Stages 45 pre-dialysis | No—pilot study reported. | New study-specific measure |
| Unnamed System 1[88] | Canada | In-centre HD, transplant | No—in development | PROMIS CATs (unspecified)[88] |
| *Survey data only—developers declined participation in optional FU interview | | | | |
| 'Your symptoms matter'[41 89] | Canada | In-centre HD | Yes—pilot study reported | ESAS-r: Renal |
| Ambuflex (PRO-KID Non-inferiority pragmatic randomised controlled trial evaluating incorporating Ambuflex platform)[24 62 90–93] | Denmark | **Ambuflex** CKD Stages 45 pre dialysis, Conservative Care **(PRO-KID** CKD Stages 45 pre dialysis, home) | **Ambuflex**—Yes **(PRO-KID**—trial in progress, | **Ambuflex** – 27 item Renal Disease questionnaire[90] **(PRO-KID** 1. Renal-specific domains, items from – KDQOL-SF – EORTC – SF-GH1 2. Additional research PROs[24] |
| SMaRRT-HD[20 43] | USA | In-centre HD | Unknown—feasibility study reported | SMaRRT-HD —Study-specific measure |
| **System identified only—no survey or interview data included in evidence synthesis | | | | |
| 'Derby Evaluation of Illness'[72] | UK | CKD stages 4/5 pre-dialysis, in-centre HD, PD | Unknown—feasibility study reported | Six separate domains assessed by VAS: general well-being, pain, sleep, breathing, energy, fistula function and appetite. |
| EMPATHY study[22 60 67 94] | Canada | In-centre HD | Yes—Cluster RCT in progress | ESAS-r: Renal/IPOS-Renal and/or the EQ-5D-5L |
| K-Pal[19] | USA | Patient's ≥60 years of age with ESRD on HD | Unknown—feasibility study reported | Short-Form McGill Pain Questionnaire 2 (SF-MPQ-2), Patient Health Questionnaire-9 (PHQ-9), Generalized Anxiety Disorder 7 Item Survey (GAD-7), Dialysis Symptom Index (DSI), KDQOL-36 |
| eNephro[44] | France | CKD stage 3B/4, stage 5D CKD on dialysis (PD/HD), Transplant | Unknown—Pragmatic RCT (anticipated date of study completion December 2018 NCTO 2082093) | Symptoms, Hospitalization Anxiety Depression Scale (HADS) KDQoL 36, ReTransQoL (for transplant patients) to assess trial outcomes |
| Dutch Renal Registry[95–97] | The Netherlands | Patients undergoing dialysis | Yes—part of development of national registry of PROMs | Dialysis Symptom Index (DSI), SF-12 |

CKD, chronic kidney disease; EMPATHY, Evaluation of routinely Measured PATient reported outcomes in HemodialYsis care; EORTC, European Organisation for Research and Treatment of Cancer; ePRO, electronic patient-reported outcome; EQ5D5L, 5-level EuroQol 5 dimension questionnaire; ESRD, End Stage Renal Disease; FACIT, Functional Assessment of Chronic Illness Therapy; HD, haemodialysis; IPOS Renal, Integrated Palliative Care Outcome Scale Renal; K-Pal, iPad based ePROM application; PD, Peritoneal Dialysis; POS-S-RENAL, Palliative Care Outcome Scale - Symptoms - Renal; PROMIS, Patient-Reported Outcomes Measurement Information System; RCT, Randomised Controlled Trial; RePROM, The use of an electronic Patient-Reported Outcome Measure in the Management of Patients with Advanced Chronic Kidney Disease (CKD); SF-12, Short-form 12 item; SF-GH1, Short form 36_Global Health1; SMaRRT-HD, Symptom Monitoring on Renal Replacement Therapy-Hemodialysis; SONG HD, Standardised Outcomes in Nephrology Haemodialysis; VAS, Visual Analogue Scale.

**Table 2** System characteristics

| System characteristic/feature *multiple response options total ≠ 100% | Number n=10 (skipped response) | Response % |
|---|---|---|
| System launched | (0) | |
| In development | 1 | 10% |
| <1 year | 1 | 10% |
| 1–5 years | 5 | 50% |
| >5 years | 3 | 30% |
| Primary objective of system* | (0) | |
| Improving symptom assoc with CKD | 7 | 70% |
| Improving symptom assoc with treatment | 5 | 50% |
| Psychosocial care | 0 | 0% |
| Facilitate communication | 4 | 40% |
| Research | 3 | 30% |
| Benchmarking | 0 | 0% |
| Commissioning | 0 | 0% |
| Support transition of care | 0 | 0% |
| Secondary objective of system* | (0) | |
| Improving symptom assoc CKD | 2 | 20% |
| Improving symptom assoc treatment | 4 | 40% |
| Psychosocial care | 8 | 80% |
| Facilitate communication | 6 | 60% |
| Research | 4 | 40% |
| Benchmarking | 2 | 20% |
| Commissioning | 1 | 10% |
| Support transition of care | 2 | 20% |
| Academic | (0) | |
| Charitable | 5 | 50% |
| Government | 4 | 40% |
| How developed? | 1 | 10% |
| In-house informatics team | (0) | |
| Collaboration across sectors | 1 | 10.0% |
| Commercial product | 4 | 40.0% |
| Funding source* | 3 | 30.0% |
| Commercial | 3 | 30.0% |
| Primary location of use* | (0) | |
| Primary care (community/general practice clinic) | 10.0% | 1 |
| Secondary care (hospital clinic) | 80.0% | 8 |
| Dialysis centre | 60.0% | 6 |
| Home | 20.0% | 2 |
| System platform* | (0) | |
| Non-responsive website | 2 | 20.0% |
| Responsive/mobile website | 8 | 80.0% |
| Mobile application | 2 | 20.0% |
| System access* | (0) | |
| Computer | 7 | 70.0% |
| Tablet | 10 | 100.0% |

Continued

**Table 2** Continued

| System characteristic/feature *multiple response options total ≠ 100% | Number n=10 (skipped response) | Response % |
|---|---|---|
| Interactive voice response system | 0 | 0.0% |
| Clinic-based kiosk | 2 | 20.0% |
| Smartphone | 7 | 70.0% |
| PRO selection* | (2) | |
| Automatic (by system) that is, provider | 8 | 100% |
| By patient | 0 | 0.0% |
| System security features* | (1) | |
| Secure log in | 9 | 100.0 |
| Encryption | 7 | 77.8% |
| Two factor log in | 1 | 11.1% |
| Unsure | 1 | 11.1% |
| Page features* | (1) | |
| Progress bar | 6 | 66.7% |
| Visual graphics that is, graph, diagram, chart | 4 | 44.4% |
| Automatic save function, option to save and return later | 1 | 11.1% |
| Training available | (0) | |
| Yes | 9 | 90% |
| No | 1 | 10% |
| Who receives training?* | (1) | |
| Clinical staff | 9 | 100.0% |
| Administrative staff | 4 | 44.4% |
| Patients | 7 | 77.8% |
| Carers | 0 | 0.0% |
| Form of ePRO system training* | (1) | |
| Face to face | 7 | 77.8% |
| Online | 7 | 77.8% |
| Integrated into ePRO system | 2 | 22.2% |
| Facility for patient education* | (2) | |
| Administered in system | 4 | 50.0% |
| Education linked to ePRO scores | 1 | 12.5% |
| Automatic documentation of action | 0 | 0.0% |
| No patient education offered | 4 | 50.0% |
| In development | 1 | 12.5% |
| Has the system been evaluated? | (0) | |
| Yes | 7 | 70% |
| No | 3 | 30% |

CKD, chronic kidney disease; ePRO, electronic patient-reported outcome.

process. Some interview participants were previously known to the interviewer, as experts in the field of study. Maintenance of a reflective diary, memos and field notes, along with discussions with the study management team, were used to minimise the influence of any prior relationships on data analysis.

**Table 3** Data collection and assessment

| Data collection/assessment *multiple response options total ≠ 100% | Number n=10 (skipped responses) | Response % |
|---|---|---|
| Measure development* | (0) | |
| System uses new PROM | 5 | 50 |
| System uses existing PROM | 7 | 70 |
| Timing of assessment* | (0) | |
| Prior to clinical assessment | 4 | 40.0 |
| Set time point weekly | 0 | 0.0 |
| Set time point monthly | 1 | 10.0 |
| Set time point 3 monthly | 3 | 30.0 |
| Set time point 6 monthly | 1 | 10.0 |
| Set time point annually | 0 | 0.0 |
| As required (by patient) | 2 | 20.0 |
| Other | 4 | 40.0 |
| Question format | (1) | |
| One question per page | 3 | 33.3 |
| Multiple questions per page | 0 | 0.0 |
| Mixed format of single and multiple questions | 6 | 66.7 |
| Question advancement | (1) | |
| Mouse click | 6 | 66.7 |
| Automatic on completion | 3 | 33.3 |
| Analysis metric* | (0) | |
| Change from baseline | 7 | 70.0 |
| Final value | 6 | 60.0 |
| Time to event | 0 | 0.0 |
| Other (system defined) | 2 | 20.0 |
| Avoidance of missing data* | (1) | |
| Allows multiple logins per assessment with automatic save function | 0 | 0.0 |
| Allows multiple logins per assessment with save and return later function | 3 | 33.3 |
| Allows not applicable (N/A) response | 2 | 22.2 |
| Default response pre-selected (pre-populated neutral response) | 0 | 0.0 |
| Reminders | 5 | 55.6 |
| Other: that is, mandatory fields | 4 | 44.4 |
| Notification of completed assessments* | (0) | |
| Automated submission notification to patient | 1 | 10.0 |
| Automated submission notification to clinicians | 4 | 40.0 |
| Email notification to patient from clinical team following review of responses | 2 | 20.0 |
| App-based notification | 0 | 0.0 |
| Other: none/unsure | 3 | 30.0 |
| Patient reminder format* | (0) | |
| Email | 5 | 50.0 |
| Telephone call | 2 | 20.0 |

Continued

**Table 3** Continued

| Data collection/assessment *multiple response options total ≠ 100% | Number n=10 (skipped responses) | Response % |
|---|---|---|
| Text message: SMS | 2 | 20.0 |
| Verbally | 5 | 50.0 |
| By letter | 3 | 30.0 |
| No reminder | 0 | 0.0 |
| App-based push notification | 1 | 10.0 |
| Flexible system features* | (1) | |
| Web based home and clinic login access | 7 | 77.8 |
| Multiple assessment scheduling options | 3 | 33.3 |
| Two or more sources for PRO selection (patient/provider) | 1 | 11.1 |
| Self-identification of important issues by patient (CAT functionality) | 3 | 33.3 |
| Free text availability | 8 | 88.9 |
| Multiple language availability | 3 | 33.3 |
| In app or push notifications | 0 | 0.0 |
| Other: facility for multiple lang/CAT currently not used | 1 | 11.1 |

CAT, Computerised Adaptive Testing; PROM, Patient-Reported Outcome Measure.

## Analysis

Data derived from the phase 1 review were abstracted and charted (see table 1), and descriptive statistics from the phase 2 structured survey were tabulated (see tables 2–4). Analysis of phase 3 interview data was undertaken using Codebook Thematic Analysis[33 34] and the domains from the Consolidated Framework for Implementation Research (CFIR) (figure 1). Primary data analysis was conducted by lead author (NEA) with a second investigator reviewing coding (CM) for consistency and appropriateness. The CFIR is a determinant framework that can be used to identify and delineate contextual factors (ie, barriers or facilitators) that influence the outcome of implementation efforts.[35 36] Computer-Assisted Qualitative Data Analysis Software CASDAQ (QSR NVIVO V.12[37]) was used to facilitate qualitative data analysis. Thematic analysis using a framework approach was chosen to systematically identify and analyse patterns of meaning within data, with the aim of highlighting the most salient features.[33] The CFIR was used to gain insight into the overall effectiveness of the ePRO systems and associated implementation strategies. It is possible to rate constructs using the CFIR to undertake organisational comparison via Qualitative Comparative Analysis.[38] Due to the heterogeneity of systems, their context and stage of development, there was a danger of oversimplifying complex, dynamic descriptions of implementation processes and contexts, so this aggregated data approach was not taken.

**Table 4** System reporting

| System reporting/integration *multiple response options total ≠ 100% | Number n=10 (skipped responses) | % Response |
|---|---|---|
| ePRO report access* | (0) | |
| Via clinical portal/electronic patient health record | 7 | 70.0 |
| Via immediate access that is, summary print out on completion | 4 | 40.0 |
| Results restricted to clinical encounter | 0 | 0.0 |
| Other: Registry, emailed PDF to patient | 2 | 20.0 |
| ePRO report content* | (0) | |
| Current scores including summary and individual scores | 7 | 70.0 |
| Longitudinal change | 8 | 80.0 |
| Interpretation included in report | 3 | 30.0 |
| Cut scores (eg, low, medium, high) | 4 | 40.0 |
| Population norms or reference values | 2 | 20.0 |
| Identification of meaningful change | 1 | 10.0 |
| Modifiable reports | 1 | 10.0 |
| General guidelines | 4 | 40.0 |
| Other that is, colour coding | 3 | 30.0 |
| Who has access to report/summary* | (0) | |
| Patient | 10 | 100.0 |
| Clinicians | 10 | 100.0 |
| Care provider | 3 | 30.0 |
| Multiple care provider access | 3 | 30.0 |
| Researcher | 1 | 10.0 |
| Who is responsible for initial review and action | (1) | |
| Medical staff | 4 | 44.4 |
| Nursing staff | 5 | 55.6 |
| Administrative staff | 0 | 0.0 |
| Other member of multidisciplinary team | 0 | 0.0 |
| Form of clinical response* | (0) | |
| Prescribed electronic response dependent on score | 2 | 20.0 |
| Clinician/staff follow-up; follow-up type (face to face, virtual) dependent on score/decision aids | 9 | 90.0 |
| Automatic referral to member of multidisciplinary tea | 1 | 10.0 |
| Automated patient education/message regardless of score | 1 | 10.0 |
| Self-management support resource dependent on score | 1 | 10.0 |
| Visual presentation of PROs scores* | (0) | |
| Graphical | 8 | 80.0 |
| Tabular | 3 | 30.0 |
| Numerical | 3 | 30.0 |
| Emoticon | 1 | 10.0 |
| Colour coded | 5 | 50.0 |
| | | Continued |

**Table 4** Continued

| System reporting/integration *multiple response options total ≠ 100% | Number n=10 (skipped responses) | % Response |
|---|---|---|
| Other: No score/being developed for longitudinal data | 2 | 20.0 |
| Safety alert system (eAlert)—dependent on PRO score | (0) | |
| Yes | 3 | 30.0 |
| No | 7 | 70.0 |
| Intended recipient for eAlert | (7) | |
| Clinician/staff | 3 | 100 |
| Patients | 0 | 0.0 |
| Care provider | 0 | 0.0 |
| Caregiver | 0 | 0.0 |
| Multiple recipients that is, clinicians and patients | 0 | 0.0 |
| Form of eAlert* | (7) | |
| Email | 2 | 66.7 |
| Text message/SMS | 0 | 0.0 |
| Telephone call | 0 | 0.0 |
| Verbal | 0 | 0.0 |
| Real time alert | 2 | 66.7 |
| In App notification | 0 | 0.0 |
| None | 0 | 0.0 |

ePRO, electronic patient-reported outcome.

Due to time constraints, participant checking of findings was undertaken by one participant only.

### Patient and public involvement
Patient partners were involved in the design of this study and coauthored the final manuscript; where they specifically highlighted the importance of ensuring that ePROs do not exacerbate existing health inequalities due to digital inequalities (lack of digital access or skills).

### RESULTS
The scoping review identified 14 ePRO systems across seven countries from 43 papers and one developing system identified by field experts, totalling 15 systems: Canada n=4, USA n=3, United Kingdom n=4, Denmark n=1, France n=1, The Netherlands n=1, Australia n=1 (table 1). Online supplemental file 1 includes the Preferred Reporting Items for Systematic Reviews and Meta Analyses Extension for Scoping Reviews (PRISMA ScR) diagram.

Ten system developers responded to the survey (66% response rate). System developers held varying academic and clinical academic research posts, all demonstrating outcomes methodology expertise. Key responsibilities ranged from funding acquisition and stakeholder engagement, to overseeing system development, piloting

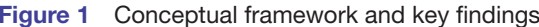

**Figure 1** Conceptual framework and key findings.

and evaluation. Seven developers agreed to a follow-up interview: one expert delegated this task to a suitable member of their study team; three declined. Reasons for not taking part in the follow-up interview were not collected. Interviews ranged from 42 to 82 min in length (mean 63 min).

### Key system characteristics

Table 2 outlines characteristics of surveyed systems and system objectives.

The most common primary objective was symptom management associated with CKD and/or renal replacement therapy by HD. However, systems were implemented across kidney disease stages and treatment modalities (see table 1 and online supplemental file 4 for additional system data). Aggregate uses of ePROs for benchmarking or commissioning of services were not common; one system collected satisfaction with care data[39] and two others assessed quality of care.[21 40] There was variation in funding source; academic (40%), government (30%) and charitable funding (30%), with 70% of systems being developed in the last 5 years. Half of the systems were developed in-house by local informatics teams, with the rest reporting collaborations across sectors with registries and technology companies. Systems allowed ePROs to be completed in different settings; the most common setting for completion was hospital clinic (80%). Only one system could be accessed by primary care.[40]

One surveyed system, Ambuflex, had been implemented into routine practice but was undergoing additional evaluation;[25] all other systems were at varying stages of study with most having conducted pilot/feasibility testing. Two identified systems were being evaluated within a cluster randomised controlled trial,[21 22] while another was being evaluated in a codesigned demonstration trial.[40] Tablet

was the most popular mode of administration. Two systems offered a clinic-based kiosk facility.[25 41] Training was available or planned for clinical staff in all systems but focused largely on system access and functionality. Seven systems had developed training materials for patients and four for administrative staff. Four systems had patient education material administered within the system, only one of these linked self-management guidelines to scores.[9] Only 60% of systems were currently linked with the EHR.

All the surveyed systems involved patients during design and implementation, from measure choice to evaluation. The ePRO Collaborate system used a codesign process to refine the dashboard and the constructs being assessed in their demonstration trial, combining the use of healthcare coproduction frameworks with stakeholder engagement.[40] Developers of the OPTimising engagement in routine collection of Electronic Patient-Reported Outcomes (OPT-ePRO) system used participatory codesign guided by normalisation process theory to guide design and planning with clinicians, patients and caregivers.[42]

Patients with CKD at varying stages were involved in the decision to create a study-specific questionnaire for the RePROM study, in order to capture outcomes they felt were most relevant.[9]

### Data collection and assessment

Table 3 summarises key data collection functions.

Similarly, half the developers reported using a newly developed PRO measure, either completely new or a mix of existing tools. Only three surveyed systems had ePRO availability in languages other than English,[20 21 41 43] although several developers reported plans for future translations, dependent on funding. There were a mixture of mechanisms to remind patients to complete

their ePROs, with email (50%) and verbal reminders (50%) being most popular. Reminders were seen as the best way to avoid missing data (55.6%), but some systems offered a save and return function.

Frequency of reporting was varied, with most common timing of assessment being prior to a clinical assessment (40%). Some systems did have set time points ranging from 1 to 6 monthly. Two systems allowed patients to complete as required.[9 41]

### System reporting

Table 4 outlines system-reporting features.

All systems allowed patients and clinicians direct access to ePRO results or summary reports. Doctors (44.4%) and nurses (55.6%) were responsible for initial review and action of results. Follow-up (face to face, virtual, telephone) was largely dependent on scores. Electronic PRO reports contained current scores (summary and individual scores) in 70% of systems, 80% presented longitudinal changes and 40% offered cut scores (low, medium and high). One system was part of an overall dashboard with the option for patients to combine ePRO scores with lab values in a graph,[40] and the Ambuflex PRO questionnaire was also viewed alongside other available clinical data, such as biochemistry, blood pressure, body weight, as a decision aid together to determine whether the patient needs a phone call or a face-to-face visit.[24]

In terms of data visualisation, graphical presentation was most popular (80%) with 50% of ePROs using colour coding to highlight significance of scores. Less frequently used approaches of score presentation were tables (30%), numerical scores (30%) and emoticons (10%). Assistance with interpretation within the report was uncommon, but 30% of ePROs had developed a safety alert system dependent on respondent scores.[9 20 25] The RePROM system produced real-time email alerts directly to the kidney nursing team,[9] while the SMaRRT HD system generated email alerts, for which the severity alert thresholds required adaption, so that they were frequent enough to ensure symptom capture without overburdening the workflow with emails (range 2–22/week).[20] The AmbuFlex system keeps track by giving patients a red, yellow and green status with non-responders presented to clinicians on an alert list.[25] The eNephro study by Thilly et al[44] was investigating the cost-effectiveness of home telemonitoring in patients with CKD. This system included a clinical decision support algorithm to detect situations of risk; relying largely on biological parameters, it could alert in cases of a complication associated with predefined symptoms.[44]

### Follow-up interviews

Semistructured interviews were undertaken with seven ePRO developers, allowing clarification of survey responses: to explore the development process and identify contextual factors working for and against implementation. All interviews took place online using videoconferencing software (Zoom) with no-one else

present. Analysis generated rich data which are presented in line with the CFIR key domains.[35] Figure 1 outlines the conceptual framework (Illustrative quotations to support the key findings are provided in online supplemental file 5).

### Intervention characteristics

Analysis highlighted the heterogeneity of the systems. However, developers agreed that systems need to be flexible, responsive, and follow an iterative development strategy. All interview participants felt that their systems would need future refinement, whether simple adaption to increase language availability or more complex programming to allow real-time symptom alerting or provision of self-management tools. Knowledge of context (population and setting) and purpose of ePRO data collection should be clearly defined and communicated to all stakeholders including patients. Developers acknowledged that the evidence of ePROs benefit is less mature in nephrology than other clinical specialities. However, participants indicated ePROs could deliver a relative advantage over current care delivery: improving communication, supporting symptom management, enhancing self-care and shared decision making, with potential to reduce healthcare utilisation. However, all participants recognised the challenges, particularly of clinician engagement. They cited clinician anxieties around potential increases in workload, attendance to symptoms and problems unrelated to kidney disease. Consequently, several developers chose not to include real-time safety alerts associated with severe symptom reports.

Several systems used currently available infrastructure, utilising existing platforms, registries, and validated measures. While this offered the advantage of existing expertise and reduced cost, this was offset by restrictions on local adaption and global comparability due to lack of harmonisation.

### Outer setting

All ePROs system programmes had been impacted and delayed by the COVID-19 pandemic. Several developers reported that key staff including informatics teams had reprioritised activity and had less availability to support ePRO implementation. However, they stated that the rapid shift to virtual consultations and recognition of the potential impact of ePROs in managing care, particularly outpatient planning, invigorated interest in ePROs.

Developers highlighted the importance of partnership and collaboration – across clinical, academic, and commercial organisations. Registries were seen as an ideal collection platform. However, developers indicated multi-organisational approaches could lead to competing interests and governance issues that can cause delays.

Developers acknowledged the value of integration of ePROs directly into the EHR, but not all systems had this functionality. Some still required manual entry, which was reported as time consuming, costly and would not

support real-time care. Many systems were standalone in nephrology secondary care, meaning data would not 'follow' patients across care settings and could not be used to manage transition of care needs.

## Inner setting

Developers suggested potential unintended consequences of ePRO implementation included further widening of health inequities, due to inadequate accessibility to digital devices, variable digital competence and lack of inclusivity caused by poor availability of systems in multiple languages or failure to assess cultural compatibility. Some developers were investigating such outcomes in particularly vulnerable groups, for example, elderly, ethnic minorities.

Developers cited the continued prevalence of biomedical models of care, despite general acceptance of the need to move to person-centred care, as a barrier to ePRO implementation. They indicated approaches for shifting 'power' to patients by focusing on patient understanding of ePROs and involvement at all stages of design and implementation, along with patient peer support to counteract cultural barriers.

Doctors were the professional group most often involved in review and action of ePRO responses, but several systems directed ePRO scores to nursing staff or physician assistants as the initial handler. Resources needed to be available to allow the engagement of the wider multidisciplinary team, including supportive and rehabilitation services.

While clinicians were largely receptive to ePROs, workflow compatibility was seen as crucial to uptake, and this was reflected in the preference for a focus on symptom management as primary objective of capture. Several systems included free text options to report symptoms or problems and clinicians needed reassurance that this would not significantly impact workload. Participants reported that pilot study data suggested free text responses did not adversely increase workload but provided another mechanism for patient/provider communication, likewise with initial data on use of symptom report alerts.

## Characteristics of individuals

Participants discussed the importance of measuring outcomes, which matter most to patients but emphasised the hypothetical danger of over digitising the patient experience, meaning that clinicians risk responding to scores alone rather than listening and connecting with the patient to further interpret ePRO data. Several participants looked forward to a future when ePROs might be given similar status to other forms of medical data. They noted clinicians frequently discuss burden associated with the collection and use of ePROs but did not consider similar factors associated with non-essential blood or invasive tests.

Developers reflected on the varied ability to deal effectively with responses, both at an organisational and individual level. They recognised the heavy and complex workload within the specialty, and while some systems offered management guides or further resources, participants were not all in favour of prescriptive treatment decision algorithms, some preferring to allow the exercise of clinical judgement.

## The process of implementation

Systems were in different stages of development and use, with some being part of wider research or quality improvement programmes. A pragmatic approach to implementation was often described: for example, measures were chosen according to what was already being used and what patients would be prepared to complete. Existing platforms that had been previously assessed for governance and regulatory compliance were exploited. Implementation readiness for use in routine care was affected by most systems being investigated outside a real-world context. Sustainability was an issue: future planning would need to include activities undertaken by research personnel, including administration tasks, completion support and training. Long-term IT and informatics support needs to be factored into costing for maintenance and adaption.

All developers stated the importance of patient involvement at all stages and outlined different ways patients had offered input and insight into codesign/production to research participation in usability testing. Staff dedicated to engagement were useful, ensuring extended community groups were involved.

Education and training associated with the knowledge needed to support ePROs appeared to be underdeveloped when compared with other aspects of system design. Training resources were often in development and current iterations focused predominantly on system use and did not extend to interpreting or actioning responses. It was recognised that training was time-consuming and needed to be ongoing.

## Overarching themes: barriers and facilitators to effective implementation

Across all CFIR constructs, analysis identified potential barriers and facilitators. Barriers included the culture associated with biomedical models, variable clinical engagement particularly from doctors and governance issues. Participants suggested facilitators included the development of patient-driven systems, utilisation of clinical champions, who need both time and belief to promote the system, and adapting existing resources.

## DISCUSSION

This review summarises and integrates survey and interview data on the features and implementation factors of 15 nephrology ePRO systems used across seven countries with table 5, outlining key priorities for successful implementation.

The majority of surveyed systems prioritised symptom management over other uses of ePROs such as benchmarking and commissioning (see table 2). While some

**Table 5**  Key priorities for successful implementation

| | |
|---|---|
| Macro level priorities (National)—formation of national collaborative groups to promote equitable and sustainable uptake of ePROs systems that have multiple applications including support from professional bodies | Harmonisation of collection platforms and governance systems for example, National Registry—allowing data linkage to other healthcare settings that is, primary care<br>Harmonisation of measures (including item banks)<br>Harmonisation of methods of interpretation and analysis metrics<br>High-quality research to demonstrate empirical evidence of benefit including real-world evaluation<br>Continued research on psychometric properties and interpretation of measures<br>Investigation of new technologies, ie, CAT<br>Endorsement of key guidelines such as the PCORI Users' Guide to Integrating Patient-Reported Outcomes in Electronic Health Records, [98] ISOQOL Implementing PROs in clinical practice [99] and the ISPOR Validation of Electronic Systems to Collect Patient-Reported Outcome (PRO) Data Recommendations.[100]<br>Provision of adequate resources |
| Meso level priorities (organisational)—key contributors to include clinicians, patients, carers, IT, informatics, QA, governance departments—local systems specificity to maximise implementation and minimise effects on workflow | Involve all stakeholders—define and communicate key system objectives<br>Use flexible system design to facilitate local, regional and national compatibility<br>Support localised adaption that is, clinic level and resourcing EHR compatibility<br>Allow varying modes of administration, including paper<br>Reminders—varying forms (email, app alert, etc.)<br>Alert systems—develop pathways and algorithms for action that compliment workflow<br>High levels of automation to reduce workload<br>Data collection features—need efficient, easy user experience<br>Easy to access, save, submit and review<br>Optimise data interpretation and visualisation<br>Infrastructure support to deliver 'holistic care'—treatment and care algorithms<br>Appointment and support for clinical 'champions'<br>Patient peer support via 'Patient Navigators' Training packages to support implementations for all users<br>Sustained IT support<br>Ongoing evaluation programmes<br>Localised education on the management of long-term conditions |
| Micro level priorities (individual) involve all key stakeholders and undertake meaningful PPI activity to ensure ePROs are accessible and inclusive for population | Identify local needs that is, language availability requirements<br>Provide support for those with poor health or digital literacy (need an assessment of needs)<br>Consider multi-media PROMs<br>Optimise accessibility—allow onsite completion via kiosk or tablet, enable 'bring your own device'<br>Evaluation by patients to ensure systems are not over-digitising the patient experience by not discussing responses with individuals<br>Meaningful follow-up of PROs responses, tailored to individual needs |

CAT, Computerised Adaptive Testing; EHR, electronic health record; ISOQOL, International Society for Quality of Life Research; ISPOR, The International Society for Pharmacoeconomics and Outcomes Research; PCORI, Patient-Centred Outcomes Research Institute; PPI, patient and public involvement.

systems were designed to collect ePROs across CKD stages and treatment modalities, patients undergoing HD were the most common population for ePROs use. According to recent reviews, patients with CKD report between 56 and 68 common and/or severe signs and symptoms, with the exact symptom burden often determined by stage of disease and treatment.[3 45] Participants in this study reflected that symptom management was a good primary objective on which to concentrate early implementation efforts, with capability to add additional functionality and use.

None of the surveyed systems was reported as being developed to align with national quality measures, such as two times yearly In-Center Hemodialysis Consumer Assessment of Healthcare Providers and Systems[46] and annual assessment of HRQOL using the KDQOL-36 included in the U.S. Medicare End Stage Renal Disease Quality Incentive Program.[47] However, increased utilisation of national registries to collect ePROs for routine care and population level use[48] highlights the potential to assess HRQoL and symptoms through a quality registry.

## Stakeholder involvement

This study highlights the increased development of nephrology ePROs in the last 5 years, with all developers reporting positive and negative effects of the COVID-19 pandemic. While there were delays to implementation due to reprioritisation of activity, the increased drive to deliver patient centred care via telemedicine has enhanced attention on ePROs.

Patient and public involvement (PPI) is a prerequisite to inclusive and equitable PROs' research[49] and the meaningful involvement of patients in ePROs implementation programmes is crucial to effective uptake.[50] Well-established PPI groups, reflective of the total population, can offer advice on measure choice, language requirements and support resources. Patient input in design of software features including reporting and data visualisation can ensure data is interpreted correctly[51] using various devices such as graphs, emoticons and heat maps.[52 53] Patient 'navigators' can offer health literacy and communication peer support for successful digital completion.[54] All the surveyed systems had involved patients in aspects of development from measure choice to evaluation.

While other important contributors to ePROs implementation are quality assurance, governance, informatics and information technology departments, any successful ePRO system will need effective clinician 'buy in'. This study identified this as a key challenge, particularly from those clinicians not involved in ePROs research. Clinical staff state they are anxious about disruptions to already tight workflows and while ePROs are seen as a useful conversation starter, clinicians fear ePROs lack clinical utility for decision-making, citing the current lack of empirical evidence on the benefits of using this data in nephrology.[32 55] Indicating a need for research to improve the evidence base of ePROs use at individual and aggregated levels,[56] including studies which employ cluster designs and use techniques to maintain allocation concealment.[57] Studies included in this review are going some way to provide this evidence.

Developers described utilising the assistance of 'clinician champions'; individuals with both the time and influence to drive ePRO implementation. These champions would encourage clinicians to view ePRO scores as key health data, in the same way, as a biomarker or a lab test, to manage care.

Studies have highlighted tension from clinicians in their ability to respond to ePROs, whether due to workload or belief that their scope of practice should be confined to matters related to nephrology.[32 58 59] Some clinicians indicate that they are more comfortable assessing, rather than actively managing, psychosocial symptoms.[41] While holistic care is advocated within kidney care, it is not necessarily supported within the system, and multidisciplinary approaches and further training in managing mental health problems are needed.[60] Although research from other specialties suggests patient encounters using ePROs do not take longer,[61] studies on workflow changes and impact are required. Training in all systems was predominantly linked to ePROs system functionality; formal interpretation guidance beyond provision of cut scores or basic colour coding was rare or 'to be developed'. Electronic questionnaires which use and calculate scores or colour codes specifically for decision aids for treatment are considered medical devices and must comply with relevant legislation.[62] Most systems offered or were developing self-management guidance for patients. Effective training need analysis at the planning stage is crucial to assess the support needed by the multidisciplinary team to respond to ePROs. In general, systems were being developed iteratively, with training and support materials being developed or expanded after launch and this may be counterproductive to early and sustained engagement from clinical teams.

Further research is needed to investigate how we interpret PROs for CKD management and event prediction, such as dialysis start time, the impact of changing dialysis modality or the decision to undertake a conservative care pathway.

## Real-world implementation

A shift to implementation science for guidance on how to promote adoption, enhance readiness and optimise use of ePROs in real-world settings will also identify any unintended consequences of use, such as exacerbation of health inequalities and potential over digitisation of the patient experience.[63 64]

Many groups who are already subjected to disadvantage and worse health outcomes are also subjected to digital exclusion.[65] Flexible systems enhance accessibility, offering patients' choice on time, frequency, mode of administration and place of completion. Electronic PROs are commonly completed in a healthcare facility via tablet, where completion support is available. However, onsite completion may impact workflow and support assistance traditionally delivered by research staff will need to be factored into sustainability plans. Home completion may need to be incentivised and alternative means of completion should be offered as required, that is, voice activation response systems, hardcopy by post, following assessment of local population needs. Combinations of digital inclusion approaches are likely needed, to support patients with the digital and health literacy skills required to negotiate both their EHR and any corresponding ePRO data[66] and offer access to hardware.[49 65]

Early findings from the EMPATHY study stress the difficulty in implementing ePROs in HD settings, indicating that routine PROs use failed to demonstrate a significant improvement in patient–clinician communication.[67] While these findings suggested relatively good communication pretrial, the qualitative data offered potential reasons for no effect: insufficient patient and clinician understanding of the purpose of PROs, challenges with administration, inconsistencies with PROs as communication tools and limited perceived value. Highlighting that evaluation of current ePROs systems should be investigating implementation outcomes (ie, fidelity, appropriateness, acceptability, feasibility, reach, adoption and sustainability)[20 40] as well as whether ePROs are excluding already disadvantaged groups.[68] Evaluation using implementation science methodology, while formally investigating the effectiveness of ePROs interventions, offers a potential avenue to build the evidence base on benefit while supporting adoption, adaption and sustainability.[69]

While effort was made to comprehensively identify nephrology ePRO systems for inclusion, a key methodological limitation was the possibility that systems which were not referenced in the literature, available in English, would be missed. This method placed reliance on appropriate and reliable indexing systems. However, consultation with experts in the field, including study participants, to identify suitable systems aimed to minimise these effects. It is suggested that as ePROs systems proliferate in healthcare settings, an international prospective register of clinical ePROs systems could offer one solution for knowledge transfer and collaboration, but issues of ownership, resource and upkeep would need to be addressed. Additionally, while the sample sizes were small, sampling

was purposive, with cases being selected by virtue of their capacity to provide richly textured information, relevant to the phenomenon under investigation.[70] The prevailing concept for sample size in qualitative research is 'saturation'; however, this study has been guided by the concept of 'information power', that is, the larger information power the sample holds, the lower N is needed.[71] The specificity of experiences and knowledge offered by the participants for an in-depth analysis meant the actual sample held adequate information power to develop new knowledge, referring to the aim of this study.

### Areas for future research

Data linkage is central to multiple usages and harmonisation of collection platforms and associated measures will support this. Integration of ePROs within the EHR can offer 'one stop' access for clinicians and patients, if given access. Extended accessibility across healthcare providers would allow long-term monitoring across the kidney care continuum. A study of daily ePROs collection reported specific events such as fistula formation and modality changes led to PRO changes, demonstrating PROs can capture differential patient experience in CKD.[72] This study highlights the need to both implement measures that are sensitive and responsive enough to detect clinically relevant change, such as disease progression, over both short and long term, while considering how best to minimise respondent burden.

New technologies offer a potential solution, computerised adaptive tests (CATs) offer sophisticated measure delivery using algorithms based on item response theory to personalise ePROs for patients. CAT-based measures are shorter, more accurate and efficient.[73] None of the reviewed systems was currently using CAT administration; however, several developers reported that this was where they saw the future of ePROs and were actively researching this area; a renal-specific item bank and associated CAT is currently being developed for use in the UK.[74]

To conclude, while there has been considerable research investment in the development of ePROs, to measure CKD symptoms and HRQOL, the next step is to accelerate the implementation gap between research and practice; this study supports this objective by outlining key system features and exploring factors to optimise the delivery ePROs in routine care settings within nephrology.

### Author affiliations

[1]Institute of Applied Heath Research, Centre for Patient Reported Outcomes Research, University of Birmingham, Birmingham, UK
[2]Research, Development and Innovation, University Hospitals Birmingham NHS Foundation Trust, Birmingham, UK
[3]NIHR Applied Research Collaboration, West Midlands, University of Birmingham, Birmingham, UK
[4]School of Allied Health and Community, University of Worcester, Worcester, UK
[5]NIHR SRMRC, University Hospitals Birmingham NHS Foundation Trust and University of Birmingham, Birmingham, UK
[6]NIHR Blood and Transplant Research Unit (BTRU) in Precision Transplant and Cellular Therapeutics, University of Birmingham, Birmingham, UK
[7]Department of Renal Medicine, University Hospitals Birmingham NHS Foundation Trust, Birmingham, UK
[8]NIHR Birmingham Biomedical Research Centre, University of Birmingham and University Hospitals Birmingham NHS Foundation Trust, University of Birmingham, Birmingham, UK
[9]Birmingham Health Partners Centre for Regulatory Science and Innovation, University of Birmingham, Birmingham, UK
[10]Patient Partner, Institute of Applied Health Research,Centre for Patient-Reported Outcomes Research (CPROR), University of Birmingham, Birmingham, UK

**Contributors** NEA, MC, CM, PC, OLA and DK were involved in the conception of the study, and methodological design. NEA conducted data acquisition. NEA and CM undertook analysis and interpretation. NEA wrote the first and revised drafts of the manuscript. RV is a Patient Partner, involved in study design and review of key study documents including protocol, participant information sheet, survey and topic guide. All authors were involved in the final approval of the version to be submitted and published, and agree to be accountable for all aspects of the work in ensuring that questions related to the accuracy or integrity of any part of the work were appropriately investigated and resolved. NEA is responsible for the overall content as the guarantor.

**Funding** This manuscript refers to independent research funded by the National Institute for Health and Care Research (NIHR) under its Clinical Doctoral Research Fellowship Programme (Grant Reference ICA-CDRF-2018-04-ST2-027).

**Competing interests** NEA receives funding from the National Institute of Health and Care Research (NIHR) under its Clinical Doctoral Research Fellowship Programme (Grant Reference ICA-CDRF-2018-04-ST2-027), NIHR Applied Research Collaboration (ARC), West Midlands and declares personal fees from GlaxoSmithKline (GSK) outside the submitted work. MC is Director of the Birmingham Health Partners Centre for Regulatory Science and Innovation, Director of the Centre for the Centre for Patient Reported Outcomes Research and is a National Institute for Health and Care Research (NIHR) Senior Investigator. MC receives funding from the NIHR Birmingham Biomedical Research Centre, NIHR Surgical Reconstruction and Microbiology Research Centre, NIHR Blood and Transplant Research Unit (BTRU) in Precision Transplant and Cellular Therapeutics, and NIHR ARC West Midlands at the University of Birmingham and University Hospitals Birmingham NHS Foundation Trust, Health Data Research UK, Innovate UK (part of UK Research and Innovation), Macmillan Cancer Support, SPINE UK, UKRI, UCB Pharma, Janssen, GSK and Gilead. MC has received personal fees from Astellas, Aparito Ltd, CIS Oncology, Takeda, Merck, Daiichi Sankyo, Glaukos, GSK and the Patient-Centered Outcomes Research Institute (PCORI) outside the submitted work. DK reports grants from Macmillan Cancer Support, Innovate UK, the NIHR, NIHR Birmingham Biomedical Research Centre, and NIHR SRMRC at the University of Birmingham and University Hospitals Birmingham NHS Foundation Trust, and personal fees from Merck and GSK outside the submitted work. DK has received funding from the NIHR and Kidney Research UK and is Chief Investigator for the RePROM and RCAT studies. CM receives funding from the National Institute for Health Research (NIHR) Surgical Reconstruction and Microbiology Research Centre, the NIHR Birmingham-Oxford Blood and Transplant Research Unit (BTRU) in Precision Transplant and Cellular Therapeutics, Innovate UK, and has received personal fees from Aparito Ltd outside the submitted work. OLA receives funding from the NIHR Birmingham Biomedical Research Centre (BRC), NIHR Applied Research Collaboration (ARC), West Midlands, NIHR Birmingham-Oxford Blood and Transplant Research Unit (BTRU) in Precision Transplant and Cellular Therapeutics at the University of Birmingham and University Hospitals Birmingham NHS Foundation, Innovate UK (part of UK Research and Innovation), Gilead Sciences Ltd, Janssen Pharmaceuticals, Inc., and Sarcoma UK. OLA declares personal fees from Gilead Sciences Ltd, GlaxoSmithKline (GSK) and Merck outside the submitted work. The views expressed are those of the author(s) and not necessarily those of the NHS, the NIHR or the Department of Health and Social Care. The study sponsor and funders have no role in study design, including collection, management, analysis, and interpretation of data; writing of the report and the decision to submit the report for publication.

**Patient and public involvement** Patients and/or the public were involved in the design, or conduct, or reporting, or dissemination plans of this research. Refer to the Study design section for further details.

**Patient consent for publication** Not applicable.

**Ethics approval** This study involves human participants and was approved by Interviews with professional experts—Ethics approval was issued on 06/07/2021

by University of Birmingham Science, Technology, Engineering and Mathematics Ethical Review Committee: ERN_21-0495.

**Provenance and peer review** Not commissioned; externally peer reviewed.

**Data availability statement** Data are available upon reasonable request. De identified data may be available upon reasonable request via the corresponding author (ORCID 0000-0002-0614-3198).

**ORCID iDs**
Nicola Elizabeth Anderson http://orcid.org/0000-0002-0614-3198
Derek Kyte http://orcid.org/0000-0002-7679-6741
Christel McMullan http://orcid.org/0000-0002-0878-1513
Olalekan Lee Aiyegbusi http://orcid.org/0000-0001-9122-8251
Melanie Calvert http://orcid.org/0000-0002-1856-837X

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
