## [Reviewer comments · BMJ Open]

ARTICLE DETAILS

TITLE (PROVISIONAL)	Global Use of Electronic Patient-Reported Outcome Systems in Nephrology: A Mixed-Methods Study
AUTHORS	Anderson, Nicola; Kyte, Derek; McMullan, Christel; Cockwell, Paul; Aiyegbusi, Olalekan Lee; Verdi, Rav; Calvert, Melanie

VERSION 1 – REVIEW

REVIEWER	Guerraoui, Abdallah Groupement Hospitalier Edouard Herriot, Caly dial Service de Néphrologie
REVIEW RETURNED	30-Jan-2023

GENERAL COMMENTS	Thank you for giving me the opportunity to read this manuscript. The patient-reported measures (PROs) are important for person-centred care. PROs remain difficult to use in routine. This work gives hope to use PROs in clinical practice and not only for research. The objective of the study is clear. The study design is appropriate to answer the research question. The statistics used are appropriate and described fully. The results of the research are presented clearly and answers of question/objective. I approve the publication of this article without recommendation
--

REVIEWER	Oberdhan, Dorothee Otsuka Pharmaceutical Development and Commercialization Inc
REVIEW RETURNED	21-Feb-2023

GENERAL COMMENTS	The authors have provided a good overview of the published landscape of ePRO use in nephrology. Some points to enhance the manuscript: 1) Table 1: size of the system implementation - country, regional, health system, clinic, etc level? Frequency of assessments? Number of patients covered? (some of this is covered in other tables but I can't connect the information to systems)2) How did the included ePRO measures line up with national quality measures?3) It would be useful to see a mapping of assessed concepts across the selected ePRO systems. Am I correct in assuming that no system included patients in the selection of concepts of interest and all selection was clinician/organization driven? The way I read it patients were involved in the system implementation to some degree - but what about the content? Interview/survey questions give the impression that "validation" relates much more to psychometrics but content validity for the target population is not addressed well. We can expect that certain symptoms are more reflective of the patient experience related to dialysis, others with
--

	ESKD, while another set of symptoms might be reflective of the patient experience in early kidney disease. Do any systems adapt ePRO measures based on the patient's disease progression? If not that might be an unmet need for future implementations to be more effectively following patients over time. 4) Comment further on the training provided to clinicians with interpreting PRO scores, the meaningfulness and relevance to practice and patient outcomes.
--	--

REVIEWER	Bouts, Antonia University of Amsterdam
REVIEW RETURNED	26-Feb-2023

GENERAL COMMENTS	The aim of this study is to identify existing and developing ePRO systems in nephrology ascertaining factors for successful implementation. The manuscript does not read well and easy, does not describe clearly how they identified existing ePRO systems and missed existing ePRO systems in the "search". It is not clear to me how they did their search. Results are descriptive, extremely long and I do not get the factors that guarantee successful implementation which is the aim of this study. I miss a good conclusion in the Discussion.
--

VERSION 1 – AUTHOR RESPONSE

Reviewer 1 comments

Thank you for giving me the opportunity to read this manuscript. The patient-reported measures (PROs) are important for person-centred care. PROs remain difficult to use in routine. This work gives hope to use PROs in clinical practice and not only for research.

The objective of the study is clear. The study design is appropriate to answer the research question. The statistics used are appropriate and described fully. The results of the research are presented clearly and answers of question/objective.

I approve the publication of this article without recommendation

Thank you for taking the time to review this manuscript, your comments are appreciated.

Reviewer 2 comments

The authors have provided a good overview of the published landscape of ePRO use in nephrology.

Some points to enhance the manuscript:

1) Table 1: size of the system implementation - country, regional, health system, clinic, etc level? Frequency of assessments? Number of patients covered? (some of this is covered in other tables but I can't connect the information to systems)

Thank you for taking the time to review this manuscript and the suggestions to enhance the paper. As suggested, additional data has been added to table 1 in order to provide further context of the reviewed ePRO systems. However, given the size of the table with the additional data exceeds 2 pages, the original table has been kept in the main document as per BMJ Open guidelines, with reference to supplementary file 4 (table with additional data) on pages 9, 11, 16.

2) How did the included ePRO measures line up with national quality measures?

Thank you, the below text has been added to page 31, lines 450 to 456.

None of the surveyed systems were reported as being developed to align with national quality measures, such as twice yearly In-Center Hemodialysis Consumer Assessment of Healthcare Providers and Systems (ICH CAHPS) (1) and annual assessment of HRQOL using the KDQOL-36 included in the U.S. Medicare End Stage Renal Disease Quality Incentive Program (2). However, the increased use of national registries to collect ePROs for routine care as described and annually at population level (3) highlights the potential to assess HRQoL and symptoms through a quality registry.

3) It would be useful to see a mapping of assessed concepts across the selected ePRO systems

Thank you, yes, it is agreed this would be useful; due to word limits in the main manuscript this data has been added to table in supplementary file 4 outlining the ePROs systems, their selected measures and the domains/items that are assessed. Reference to supplementary file 4 has been added on pages 9,11,16.

4) Am I correct in assuming that no system included patients in the selection of concepts of interest and all selection was clinician/organization driven? The way I read it patients were involved in the system implementation to some degree - but what about the content? Interview/survey questions give the impression that "validation" relates much more to psychometrics but content validity for the target population is not addressed well.

Thank you for pointing this out so that the manuscript could be clarified – all the systems did involve patients in measure choice/content either as part of their patient and public involvement initiatives or a formal research study on measure choice, for example in the RePROM study it was patient input which led to the decision to create a study specific measure. The manuscript has been updated to illustrate this, see page 18 lines 261-263.

5) We can expect that certain symptoms are more reflective of the patient experience related to dialysis, others with ESKD, while another set of symptoms might be reflective of the patient experience in early kidney disease. Do any systems adapt ePRO measures based on the patient's disease progression? If not that might be an unmet need for future implementations to be more effectively following patients over time.

Thank you, yes, we would agree we have discussed the potential role of CATs but additionally have added the below text to the manuscript; page 36, line 570 to 577

Extended accessibility across healthcare providers would allow long-term monitoring across the kidney care continuum. A study of daily ePROs reported specific events such as fistula formation and modality changes led to PRO changes demonstrating PROs can capture differential patient experience in CKD (4). This study highlights the need to both implement measures that are sensitive and responsive enough to detect clinically relevant change over both short and long term, whilst considering how best to minimise respondent burden.

6) Comment further on the training provided to clinicians with interpreting PRO scores, the meaningfulness and relevance to practice and patient outcomes.

Further comment below provided on page 33, lines 498 to 505, lines 510-512

Training in all systems was predominantly linked to ePROs system functionality; formal interpretation guidance beyond provision of cut scores or colour coding was rare or 'to be developed'. Electronic

questionnaires which use and calculate scores or colour codes specifically for decision aids for treatment are considered medical devices, and must comply with relevant legislation.....Further research is needed to investigate how we interpret PROs for CKD management and event prediction; such as dialysis start time, the impact of changing dialysis modality, or the decision to undertake a conservative care pathway.

Reviewer 3 comments

The aim of this study is to identify existing and developing ePRO systems in nephrology ascertaining factors for succesful implementation. The manuscript does not read well and easy, does not describe clearly how they identified existing ePRO systems and missed existing ePRO systems in the "search". It is not clear to me how they did their search. Results are descriptive, extremely long and I do not get the factors that garantee succesful implementation which is the aim of this study. I miss a good conclusion in the Discussion.

Thank you for your comments. Revisions have been made to provide further clarity (see tracked changes). To assist in setting out the principal factors that support successful implementation we had included in the original manuscript **Box 1 Key priorities for succesful implementation**, which set out key factors at macro, meso and micro population levels, derived from our analysis of the qualitative data using the determinant framework the Consolidated Framework for Implementation Research (see page 29-30). We cannot guarantee that addressing these factors will lead to successful implementation but aimed to highlight those matters for consideration. This box has been moved to the beginning of the discussion.

We have included more details on our search of the published literature see page 8 lines 147 to 154 and the Supplementary file 1 for the full search strategy and PRISMA ScR diagram. We have also outlined more clearly the inclusion/exclusion criteria (see page 7 lines 134 to 141) which could be a reason why it was felt that systems had been missed. For example, we chose not to include systems designed for paediatric CKD populations due to additional complexity associated with proxy measures, age appropriate measures, outcomes reported in children which are not commonly reported in adults i.e., ‘feeling left out’ and argue such systems would require a separate study.

We have also included a section on the difficulties of identifying ePRO systems which have not been published in the literature, in the discussion section where we discuss the limitations of our methodology: page 35-36 lines 551 to 556 of the discussion.

Thank you again for taking the time to consider this newly revised manuscript. Having fully addressed all the reviewers and editors comments, the word count has increased to 5048. This paper reports a mixed methods study, with both survey and qualitative study components (methods, analysis and results) which have been integrated into overall findings, which has contributed to this word count.

VERSION 2 – REVIEW

REVIEWER	Oberdhan, Dorothee Otsuka Pharmaceutical Development and Commercialization Inc
REVIEW RETURNED	21-May-2023
GENERAL COMMENTS	Thank you for providing the revised manuscript and addressing my questions/comments. I have no additional comments.